# Integrated into Environmental Biofilm *Chromobacterium vaccinii* Survives Winter with Support of Bacterial Community

**DOI:** 10.3390/microorganisms8111696

**Published:** 2020-10-30

**Authors:** Daria A. Egorova, Olga L. Voronina, Andrey I. Solovyev, Marina S. Kunda, Ekaterina I. Aksenova, Natalia N. Ryzhova, Ksenya V. Danilova, Valentina S. Rykova, Anastasya A. Scherbakova, Andrey N. Semenov, Nikita B. Polyakov, Daniil A. Grumov, Natalia V. Shevlyagina, Inna V. Dolzhikova, Yulia M. Romanova, Alexander L. Gintsburg

**Affiliations:** 1N.F. Gamaleya National Research Center of Epidemiology and Microbiology, Ministry of Health, 123098 Moscow, Russia; dronnias@gmail.com (A.I.S.); markunda99@gmail.com (M.S.K.); aksenova16@yandex.ru (E.I.A.); rynatalia@yandex.ru (N.N.R.); kseniya.2494.danilova@gmail.com (K.V.D.); valentinarycova@inbox.ru (V.S.R.); nas.scherbakova99@bk.ru (A.A.S.); semenov_an91@mail.ru (A.N.S.); polyakovnb@gmail.com (N.B.P.); grumovda@gmail.com (D.A.G.); nataly-123@list.ru (N.V.S.); i.dolzhikova@gmail.com (I.V.D.); genes2007@yandex.ru (Y.M.R.); gintsburg@gamaleya.org (A.L.G.); 2Vernadsky Institute of Geochemistry and Analytical Chemistry of Russian Academy of Sciences, 119991 Moscow, Russia; 3Sechenov First Moscow State Medical University of the Ministry of Health of the Russian Federation, 119991 Moscow, Russia

**Keywords:** *Chromobacterium vaccinii*, biofilm, sharing goods, social cheater, bog microbiome, bacterial genome, violacein, cold adaptation, IDBac, QS mutant

## Abstract

*Chromobacterium* species are common in tropical and subtropical zones in environmental samples according to numerous studies. Here, we describe an environmental case of resident *Chromobacterium vaccinii* in biofilms associated with *Carex* spp. roots in Moscow region, Russia (warm-summer humid continental climate zone). We performed broad characterization of individual properties as well as surrounding context for better understanding of the premise of *C. vaccinii* survival during the winter season. Genome properties of isolated strains propose some insights into adaptation to habit and biofilm mode of life, including social cheaters carrying *ΔluxR* mutation. Isolated *C. vaccinii* differs from previously described strains in some biochemical properties and some basic characteristics like fatty acid composition as well as unique genome features. Despite potential to modulate membrane fluidity and presence of several genes responsible for cold shock response, isolated *C. vaccinii* did not survive during exposure to 4 °C, while in the complex biofilm sample, it was safely preserved for at least half a year in vitro at 4 °C. The surrounding bacterial community within the same biofilm with *C. vaccinii* represented a series of psychrophilic bacterial species, which may share resistance to low temperatures with other species within biofilm and provide *C. vaccinii* an opportunity to survive during the cold winter season.

## 1. Introduction

Biofilms and microbial mats are an important part of any ecosystem and one of the main biotic factors with a dramatic impact in metabolic processes and influence on other organisms’ habits and species diversity (plants, insects, protozoans, and others). Bacterial species composition of biofilms both orchestrates and reflects variation in the ecological context, including pollution, climate change, other direct and indirect anthropogenic influences, as well as metabolic capacity and bioremediation processes. Despite awareness about the importance of bacterial abundance and biogeography, the worldwide distribution of bacterial species and their adaptation to different ecological niches remains poorly explored [1,2]. While clinical cases of different bacterial species isolation and description are broadly appreciated in clinical microbiology, environmental cases in microbial ecology and biogeography are limited. Reports of environmental *Chromobacterium* species isolation in Europe are rare: one in Poland (*Chromobacterium violaceum* from *Ixodes ricinus* ticks) and evidence of *C. vaccinii* isolation from bog in Tver region, Russia [3,4]. Moreover, numerous clinical cases of infections in Europe due to *Chromobacterium* species were discussed in a prism of global warming, but primary sources of infection and natural reservoirs of pathogenic *Chromobacterium* species remain undiscovered [5]. While members of the *Chromobacterium* genus are still considered as tropical/subtropical bacteria with poor viability at low temperatures, this is still questionable if changing environment and other processes affect the worldwide distribution of “tropical” *Chromobacterium* species or an abundance of these species are underestimated in the temperate climate zone. Even though some species of the genus were isolated from complex communities like rhizosphere and root-associated biofilms, most of these evidences were associated with an aquatic environment and some species were found in water samples [6,7,8,9,10,11]. Freezing water and ice coverage of water reservoirs during the winter seasons in temperate and cold climate zones might provide significant stress to *Chromobacterium* species.

Members of the *Chromobacterium* genus are known as producers of violacein, deoxyviolacein, cyanide, extracellular chitinase, and some other active compounds. These metabolites might have high environmental significance due to a broad range of biological activities, including antibacterial activity against both the planktonic and biofilm form of Gram-positive bacteria, insecticidal, antiprotozoal, possible antiviral, and fungicidal features [11,12,13,14,15,16]. Moreover, production of at least one of them, violacein, is higher in the biofilm mode of life and upregulated by intra- and interspecies quorum-sensing (QS) signals, which raises a question about the existence within multispecies biofilms in natural ecosystems [17,18]. A broad spectrum of activities provides competition advantages for *Chromobacterium* species and might promote niche partitioning in their presence, but little is known about the life of *Chromobacterium* species in non-optimal habits like climate zones with cold seasons.

Here, we describe the case of *C. vaccinii* isolation from quaking bog rhizospheres’ biofilms in the European part of Russia, provide characteristics of isolates and surrounding bacterial community, and describe a naturally occurring QS mutant proposed as an example of social exploitation of community goods and representing evolutionary pressure on social cheating within biofilm [19].

## 2. Materials and Methods

The overall workflow of the current study is presented in Figure 1.

*Samples collection and processing***.** Sediment samples, root-associated biofilms of sedge (*Carex* spp.), sphagnum moss, and water samples were collected in triplicates. For sediment samples, we scooped sediments directly into sampling plastic tubes (V = 15 mL). Water samples were collected into sampling plastic tubes (V = 15 mL) for microbiological examination and in 3-L glass bottles for water quality analysis. Sphagnum moss fragments were directly placed into sampling plastic tubes (V = 50 mL). For root-associated biofilms, we unearthed *Carex* spp. plants at the border of water and sphagnum moss, scraped rizodermis with root-associated biofilms using sterile scalpels, and then returned the plant to its initial place. All samples were immediately transferred to the laboratory at +4 °C. For dissociation of microbial aggregate, we vortexed samples at high speed for 5 min and then 10-fold diluted samples processed with conventional microbiology plating on the following solid mediums: LB, TSA, M9 salts with 1% tryptone, nutrient agar, BHI, and blood agar. Plates were incubated at 25 °C for 48 h.

*Water quality analysis* was performed in MSULab company (Moscow, Russia).

*Species identification and metabolic association network.* MALDI-TOF MS identification: we picked single colonies from solid medium and processed with MALDI-TOF MS Sample Preparation and Data Acquisition. For MALDI-TOF MS analysis, proteins were extracted by using an extended direct transfer method that included a formic acid overlay as described in [20,21]. In brief, bacterial colonies were applied as a thin film onto a MALDI ground-steel target plate (Bruker Daltonics, Billerica, MA, USA). Over each bacterial smear, 1 µL of 70% Optima™ LC/MS Grade formic acid (Fisher Chemical, Hampton, NH, USA) was added and allowed to evaporate, followed by the addition and subsequent evaporation of 1 µL of 10 mg/mL α-cyano-4-hydroxycinnamic acid solubilized in 50% acetonitrile, 2.5% trifluoroacetic acid, and 47.5% water. All solvents were HPLC or MS grade.

Measurements were performed using an UltrafleXtreme mass-spectrometer (Bruker Daltonics, Billerica, MA, USA) equipped with a smartbeam™-II laser (355 nm). Natural product spectra were recorded in positive reflectron mode (2000 shots; RepRate: 2000 Hz; delay: 8198 ns; ion source 1 voltage: 20 kV; ion source 2 voltage: 18.8 kV; lens voltage: 7.5 kV; mass range: 50 to 5000 Da, matrix suppression cutoff: 50 Da). Protein spectra were recorded in positive linear mode (1200 shots; RepRate: 1000; delay: 29,793 ns; ion source 1 voltage: 19.5 kV; ion source 2 voltage: 18.2 kV; lens voltage: 7.5 kV; mass range: 1.9 kDa to 22 kDa matrix suppression cutoff: 1.5 kDa). Protein spectra were corrected with an external Bruker Daltonics bacterial test standard (BTS). Natural products spectra were corrected with an external Bruker Daltonics peptide calibration standard and CHCA [2M + H]+ (379.0930 Da). Automated data acquisitions were performed using flexControl software v. 3.4.135.0 (Bruker Daltonics, Billerica, MA, USA) and flexAnalysis software v. 3.4. Spectra were automatically evaluated during acquisition to determine whether a spectrum was of high enough quality to retain and add to the sum of the sample acquisition.

For species identification, we used both a conventional database from Biotyper^®^ (Bruker Daltonics, Billerica, MA, USA) and a recently introduced automatic IDBac algorithm [22]. Identification was validated in a selective manner through Sanger sequencing of *16S rDNA* amplicons obtained with 27F/1294R primers.

*Sanger sequencing. 16S rDNA* amplicons obtained with 27F/1294R primers were sequenced according to the protocol of the BigDyeTerminator 3.1 Cycle Sequencing kit for the Genetic Analyzer 3500 Applied Biosystems (Waltham, MA, USA). The electrophoretic DNA separation was performed in 50-cm capillaries with POP7 polymer.

*16S rDNA bacteriome analysis.* Total DNA was isolated from 1 mL of root-associated biofilm using a ZymoBIOMICS DNA Kit (Zymo Research, Irvine, CA, USA). Fragments of 1*6SrDNA* gene containing V1-V4 hypervariable regions (max 753 bp) were amplified and used for preparation of paired-end libraries according to the KAPA HyperPlus (Roche, Basel, Switzerland) protocol. Libraries were checked with High Sensitivity DNA Chips on a 2100 Bioanalyzer System (Agilent, Santa Clara, CA, USA) and sequenced on NextSeq 500 (Illumina, San Diego, CA, USA) with NextSeq 500/550 High-Output Kit v2.5 (300 cycles). NGS data are available in GenBank: Bio Project PRJNA635774.

Two approaches were used for the data analysis. (1) The Microbial Genomics Module of CLC Genomic Workbench v.20.0.4 (QIAGEN, Germantown, MD, USA) was used with default settings to perform Operational Taxonomic Unit clustering. Greengenes database v.13_8 (Second Genome, Brisbane, CA, USA) was used as reference with a 97% threshold. (2) SPAdes v.3.13.0 (St. Petersburg genome assembler, Russia, URL: http://cab.spbu.ru/software/spades/) was used for assembling the contigs, which were identified by BLAST NCBI with the rRNA/ITS databases [23]. Coverage of contigs of the same taxonomic units was summarized and served as an indicator of the taxon abundance. Diagram was created by Microsoft Excel (Microsoft, Redmond, WA, USA).

Phylogenetic analysis for the Proteobacteria fragments of 16S ribosomal RNA gene was performed in MEGA 6.0 [24]. The evolutionary history was inferred by using the neighbor-joining method [25]. The distances for phylogeny reconstruction were computed using the Kimura 2-parameter [26], and are in the units of the number of base substitutions per site. The rate variation among sites was modeled with a gamma distribution (shape parameter = 0.25). The analysis involved 70 nucleotide sequences. All ambiguous positions were removed for each sequence pair. There were a total of 1654 positions in the final dataset. Bootstrap analyses were performed with 1000 replicates [27].

*Whole genome sequencing and annotation.* DNA from bacterial cells of pigmented GIMC1602:ChrSima_v (ChrSV) and unpigmented GIMC1601:ChrSima_w (ChrSW) strains was isolated by a Wizard Genomic DNA purification kit (Promega, Madison, WI, USA). The KAPA HyperPlus (Roche, Basel, Switzerland) protocol was used for the libraries’ preparation. Sequencing was performed on NextSeq 500 (Illumina, San Diego, CA, USA) with NextSeq 500/550 High-Output Kit v2.5 (300 cycles). CLC Genomic Workbench v.20.0.4 and SPAdes v.3.13.0 were used for genome assembling. CGView Server (http://stothard.afns.ualberta.ca/cgview_server/) was applied for the visualization of assembling results and for the genome comparison [28]. The software Rapid Annotations Subsystems Technology (RAST) and SEED were used for genome annotation [29,30]. The conserved domains of the proteins were searched complementarily by the services KEGG (http://www.genome.jp/kegg), KEGGOC (http://www.genome.jp/tools/oc), and COGs (http://www.ncbi.nlm.nih.gov/COG). Prophage sequences were revealed with the help of PHASTER (PHAge Search Tool Enhanced Release, https://phaster.ca/) [31].

WGS data are available in GenBank: Bio Project PRJNA597450. Accession Numbers of the chromosome and plasmid are CP060046 and CP060045 for GIMC1602:ChrSima_v, CP060044, and CP060043 for GIMC1601:ChrSima_w.

Whole-genome-based taxonomic analysis was made by the Type (Strain) Genome Server (TYGS), a free bioinformatics platform of DSMZ, which is available under https://tygs.dsmz.de [32].

Biosynthetic gene clusters were predicted by antiSMASH v.5.1.2 available under https://antismash.secondarymetabolites.org/ [33]. This resource allows the rapid genome-wide identification, annotation, and analysis of secondary metabolite biosynthesis gene clusters in bacterial and fungal genomes.

*Biochemical and growth properties characterization.* Biochemical properties of *C. vaccinii* isolates were characterized using NEFERMtest 24 (Erba Lachema, Brno, Czechia) and ENDOtest (Erba Lachema, Brno, Czechia). For assessment of oxidase production, we used Microbact Oxidase Strips (Oxoid, Cheshire, England) and applied a traditional protocol using colonies that were 18–24 h old. Due to bacterial pigment interferences with color development, we used two additional methods: (a) early bacterial culture (before visible pigment production—approximately 12–16 h old); (b) an additional method useful for pigmented bacterial culture: a piece of filter paper was soaked one side in bacterial culture then placed on a test strip soaked side up and removed after 3 min of exposure; this let us to remove pigmented bacteria from the strip to avoid misinterpretation of the results. For validation of this approach, we used several strains of *Pseudomonas aeruginosa* as a positive sample.

For evaluation of the pH effects on the growing properties, TSB medium was adjusted to pH in a range from 3.0 to 9.0 with step 1.0 and then inoculated with *C. vaccinii*. For evaluation of the salinity effect, TSB medium w/o NaCl was supplemented with NaCl to 0–10% with step 0.5% and then inoculated with *C. vaccinii*. We accessed growing properties through the optical density of bacterial cultures at 600 nm after 48 h of incubation.

For assessment of the viability under different temperatures, we divided the stationary phase bacterial culture into 1-mL parts and incubated at 25 °C, 4 °C, and on ice. After desired periods of incubation, serial dilutions of bacterial culture were plated on TSA and incubated at 25 °C for 48 h for CFU counting.

Mixed colony biofilm was prepared by mixing of equal amounts of overnight cultures, and 10 mkl of bacterial mixture was spotted on TSB. Plates were incubated at 25 °C for 48 h.

For analysis of membrane fluidity adaptation to different temperatures, we grew *C. vaccinii* on TSA at 25 °C during 24 h, then changed the cultivation temperature for the next 48 h, and finally proceeded to fatty acid analysis.

*Fatty acid composition and lipid A structure analysis.* Fatty acid methyl esters (FAMEs) were prepared as described previously with some modification [34]. Briefly, bacterial pellets were resuspended in solution for saponification (300 mkl) and incubated at 100 °C for 30 min. After incubation, HCl-MeOH solution (600 mkl) was added and heated at 80 °C for 10 min. Then, FAMEs were extracted by 10 min of mixing with 600 mkl Hexane-MTBE (600 mkl). The top phase was transferred into a new vial and used for analysis through GC-MS.

GC-MS analyses were carried out using Agilent 7820A (Agilent technologies, Santa Clara, CA, USA) gas chromatograph with a Maestro MS detector (Interlab, Moscow oblast, Russia) with 30 m × 0.25 mm i.d. capillary column Rxi-5ms (Restek, Bellefonte, PA, USA). The injection volume was 1 µl, with a split ratio of 10:1 splitting gas-carrier. Injector and interface temperatures were 250 and 280 °C, respectively. The temperature program for the column started at 125 °C for 0.5 min, and then rose to 280 at 5 °C/min, and then to 320 °C at 20 °C/min; the end temperature was held for 2 min. Electron impact (EI) spectra were obtained under 70 eV ionization voltage and 150 °C source temperature. Registration of spectra was performed through a full scan at 40–550 Th mass range.

Post-run analysis was performed with the following software: Agilent Mass Hunter Unknown Analysis (Agilent Technologies, Santa Clara, CA, USA), Enhanced Data Analysis (Agilent Technologies, Santa Clara, CA, USA), NIST MS Search 2.2 (NIST, Gaithersburg, MD, USA), and Microsoft Excel (Microsoft, Redmond, WA, USA). Equivalent chain-lengths of methyl ester derivatives of fatty acids were calculated as described [35].

*Electron microscopy.* For the electron microscopy, bacterial colonies were scraped off from plates, washed with sterile water three times, and fixed in 10% neutral formalin. Electron microscopy was carried out with negative staining with 1% uranyl acetate with JEM-2100 200 kV analytical electron microscope (JEOL, Tokyo, Japan).

## 3. Results

### 3.1. Quaking Bog Description

Quaking bog Volkovskoye with an abandoned quarry Sima developed over a lake approximately 7000 years ago in Moscow region, Russia (55°40′09.2″ N 36°42′44.4″ E). According to Köppen climate classification, this is a Dfb (warm-summer humid continental) climate zone. Quarry Sima was used for the exploitation of peat in 1800–1900s [36]. Nowadays, Volkovskoye is a relatively small bog—approximately 90,000 square meters in total with an open-water region less than 10,000 square meters (maximum depth 1.5 m), surrounded by fir-tree forest. The major genus of higher plants on the edge are sphagnum moss and open water *Carex* spp. The bog is outlined by a stretch of birches and pines. A floating mat of sphagnum moss has a thickness of around 40 cm. Analysis of the water chemical properties and composition confirms a low salt concentration, which is a common characteristic for quaking bogs (Table 1).

### 3.2. Individual Properties of Chromobacterium vaccinii

#### 3.2.1. Isolation and Species-Level Identification

*C. vaccinii* was found during investigation of biofilms composition in the quaking bog Volkovskoye. Biofilm samples were collected in triplicates in winter (in January 2020; average daytime temperature −7 °C (20 °F), night −13 °C (9 °F)) when the bog was covered with ice and the water temperature was 2–4 °C. While, previously, we had already found *Chromobacterium* spp. at the same bog during the summer season (August, 2019), the isolates did not survive neither at temperatures of 4–8 °C during short (few days) storage on plates with solid mediums (BHI, TSA, LB) nor during deep freezing in storage medium supplemented with glycerol (data not shown). This was the premise to collect samples during the cold season and exclude the possibility of a transient occurrence of “tropical” bacterial species carried by reservoir birds or insects during migration.

After collection and transportation of sediment samples, root-associated biofilms of *Carex* spp., sphagnum moss, and water sample bacteria were grown on the following solid mediums: LB, TSA, M9 salts with 1% tryptone, nutrient agar, BHI, and blood agar. After incubation in aerobic conditions, 25 °C for 48 h, we isolated 18 circular, smooth, and raised colonies with entire margins and deep violet pigmentation from all root-associated biofilms of *Carex* spp. but neither from water samples nor from sphagnum moss. A single CFU with the same properties was detected in one sediment sample. Identification of isolated colonies by MALDI Biotyper™ (Bruker, Billerica, MA, USA) resulted in probable genus identification with a score no more than 1.71 for *Chromobacterium* spp. At the time of identification, the number of specters in the database related to the *Chromobacterium* genus was restricted, with only two species: *C. violaceum* and *C. substugae*. Additionally, three unpigmented colonies of *Chromobacterium* spp. were found in root-associated biofilms of *Carex* spp.

To confirm they belonged to the *Chromobacterium* genus, we performed *16S rDNA* sequencing for all isolates. Identification based on *16S rDNA* sequence analysis resulted in more than 99% similarity to *C. violaceum*, *C. vaccinii*, and *C. piscinae* species. Due to the high homology between these *Chromobacterium* species, *16S rDNA* sequence analysis was not enough for secure species identification. So, we performed WGS (whole-genome sequencing) for one pigmented and one unpigmented isolate assigned as isolates GIMC1602:ChrSima_v (ChrSV) and GIMC1601:ChrSima_w (ChrSW), respectively. Assembled WGS data were used for determination of the closest type strain genome with the help of TYGS [32]. The TYGS database includes type strains of the eight *Chromobacterium* species: *C. amazonense* DSM 26508, *C. haemolyticum* DSM 19808, *C. phragmitis* IIBBL 112-1, *C. phragmitis* IIBBL 112-1, *C. pseudoviolaceum* LMG 3953, *C. sphagni* IIBBL 14B-1, *C. subtsugae* PRAA4-1, *C. vaccinii* MWU205, and *C. violaceum* ATCC 12472. The digital DNA-DNA hybridization (dDDH) value was the highest for the pairs ChrSV-*C. vaccinii* MWU205 and ChrSW-*C. vaccinii* MWU205: 87.5% and 87.4%, respectively, with confidence intervals (C.I.s) of 84.0–90.3% and 83.9–90.2% (Appendix A). The type *C. vaccinii* strain MWU205 was initially isolated from cultivated cranberry bogs in Cape Code, Massachusetts [6,7].

#### 3.2.2. General Phenotype Characterization

In order to obtain a broad description of phenotype, we performed electron microscopy imaging, biochemical characterization, growth properties, and analysis of fatty acids contents, including the lipid A structure of LPS.

Negative staining electron microscopy of pigmented and unpigmented isolates showed bacillus with a single polar flagellum for both pigmented and unpigmented isolates (Figure 2).

Biochemical characterization of isolates was different from previously described features (Table 2). Fermentation tests of all isolates were positive for gamma-glutamyltransferase, arginine dehydrolase, phosphatase, and fermentation of trehalose. Not all but most isolates were positive for N-Acetyl-beta-D-Glucosaminidase and Simmon’s citrate test. Some other infrequent biochemical features among isolates included utilization of ornithine, lysine, mannitol, xylose, arabinose, galactose, and sucrose. Isolates were able to grow in a broad a range of pH and salinity.

The most significant and valuable difference in terms of the general species description was negative oxidase reaction for all isolates. We confirmed the negative results of the oxidase test with two different approaches: using early bacterial culture (before development of pigmentation) and using the modified pigmented strain method, unpigmented isolates were also tested with both methods. Negative oxidase reaction is not common for members of Neisseriacea, but for a minor part of *Chromobacterium*, isolates were previously mentioned [37,38].

Considering the possible causes of a negative oxidase test, our genomic studies revealed that the *Chromobacterium* isolates have a versatile aerobic respiratory system, including aa3 (locus tags ChrSV_0681 -0691), cbb3 (ChrSV_4418-4423) cytochrome c oxidases, and quinol oxidases: one copy of cytochrome O ubiquinol oxidase cyo (ChrSV_1078-1082) and two copies of cytochrome d ubiquinol oxidase cyd (ChrSV_1410 -1411 and ChrSV_3076-3077). The respiratory system comprises also the complexes of enzymes responsible for NADH and succinate oxidation (ChrSV_4656-4669 and ChrSV_4524-4536). On the other hand, *Chromobacterium* is famous for cyanide production, which is a diagnostic test *of Chromobacterium* during growth on complex medium [39]. The *hcnABC* gene cluster encoding hydrogen cyanide synthase was revealed in the genomes of the pigmented and unpigmented strains (ChrSV_3735- 3737).

The oxidase test is used to identify bacteria that produce cytochrome c oxidase. However, this enzyme is cyanide sensitive, so both aa3 and cbb3 cytochrome c oxidases could be inhibited by cyanide production. The survival of *Chromobacterium* under these conditions is provided by other enzymes of the respiratory system: NADH and succinate oxidation is highly resistant to inhibition by cyanide [40]. Additionally, the ubiquinol oxidases may enhance tolerance to oxidative and nitrosative stress in some bacterial species [41].

We performed conventional fatty acid methyl ester (FAME) characterization of our isolates as the key features of microbial species characterization. The FAME profile for our type isolate ChrSV in comparison with other characterized species is presented in Table 3 and FAMEs for different isolates in this study are presented in Appendix A.

According to the FAME analysis, the predominant fatty acids (FAs) were C16: 1ω7c (43.7%), C16: 0 (28.4%), and C18: 1ω7c (12.5%). The fatty acids profile of GIMC 1602 was different from any previously reported data. The close resemblance was found in the C16 fatty acids between GIMC1602:ChrSima_v, *C. vaccinii* (MWU 300, MWU 205), *C. subtsugae* PRAA4-1, *C. sphagni* IIBBL 14B-1, and *C. haemolyticum* CCUG 53230. Some differences were found in the hydroxy fatty acids content and other minor fatty acids. Namely, GIMC1602:ChrSima_v has a low level of hydroxy fatty acid and contains 14:1 and 15:1 FA in contrast to other *C. vaccinii* strains (MWU 300, MWU 205). To determine if hydroxy fatty acid belongs to LPS or not, we analyzed the lipid A structure. Lipid A contained all three hydroxy fatty acids (Appendix A) and its structure was the same as described for *C. violaceum* NCTC 9694 [48,49].

#### 3.2.3. Genome Characterization and Comparison with Known Strains of *C. vaccinii*.

Genome analysis of the pigmented (ChrSV) and unpigmented (ChrSW) *C. vaccinii* strains demonstrated that both genomes consist of chromosome (5,278,675 and 5,273,834 bp, respectively) and plasmid (45,365 bp). Alignment of ChrSV and ChrSW chromosomes with the help of the BLAST NCBI with Genome Data Base revealed the highest homology (99.27% identity and 90% coverage) with *C. vaccinii* strain 21-1 genome, Accession Number CP017707.1. This strain was isolated from bog in Beltsville, Maryland, USA, a place with a humid subtropical climate. The chromosome of *C. vaccinii* strain 21-1 is less than the chromosome of ChrSV by 237,445 bp.

The ChrSV/ChrSW genomes have the biggest number of prophages—12 and 7 of them are intact. The genome of *C. vaccinii* strain 21-1 has only 6 prophages (4 intact), 5 of which are similar to prophages of the ChrSV/ChrSW genomes. The second *C. vaccinii* complete genome, the genome of strain XC0014 (Accession Number CP022344.1), is even smaller, and has six prophages (five intact), but only two of them are similar to prophages of the ChrSV/ChrSW genomes. It should be noted that if the first two strains were isolated from the bogs, the *C. vaccinii* XC0014 strain had another source of isolation: the soil in Zhejiang Province of China. However, the climate in Zhejiang is humid subtropical as in Beltsville, in contrast to the climate in the Moscow region.

The next difference between the ChrSV/ChrSW genomes and genomes of 21-1 and XC0014 is the presence of the plasmid (45,365 bp). In the Microbial Genome Data Base of NCBI, only 3 *Chromobacterium* genomes from 58 genomes with different levels of assembly have plasmids: *C. violaceum* FDAARGOS_635 (CP050991.1, 42,965 bp), *Chromobacterium*
*sp.* IIBBL 112-1 (NZ_CP029496.1, 17,589 bp), and *Chromobacterium**. sp.* IIBBL 274-1 (NZ_CP029555.1, 74,363 bp). The plasmid of the type strain of *C. violaceum* ATCC 12,472 was submitted in GenBank separately (MG651603.1., 44,212 bp) [50].

The ChrSV/ChrSW plasmids did not have similarity with the *Chromobacterium sp.* plasmids but were homologous to plasmids of both *C. violaceum* strains with a coverage of 83%: FDAARGOS_635-93.05% identity, ATCC 12,472-92.51%. Note that *C. violaceum* strains were isolated from different sources: FDAARGOS_635 is a clinical isolate (University of Louisville, US), and ATCC 12,472 is a freshwater isolate (Malaya, Malaysia).

So, the ChrSV/ChrSW genomes are the biggest from known *C. vaccinii* genomes and differ in the presence of a considerable number of prophages (including intact prophages) and plasmid homologous to the plasmids of *C. violaceum* strains.

### 3.3. Regulatory Nature of Unpigmented Isolate

We isolated three unpigmented strains of *C. vaccinii* and performed whole-genome sequencing for one of them: ChrSW. We identified the full *vioABCDE* operon in the genome sequence of unpigmented isolate, which gave us a reason to hypothesize on the regulatory nature of absence pigmentation in the ChrSW isolate. Regulation of violacein production depends on quorum-sensing (QS) signals [51]. Closer investigation of the genome region responsible for the LuxI/LuxR regulatory QS pathway revealed deletion of a large DNA fragment including the *luxR* sequence, which made classical positive-feedback regulation of the LuxI/LuxR system in response to AHL impossible (Figure 3).

Lack of *luxR* expression due to deletion in the genome of the ChrSW strain resulted in insensitivity to external AHL and inability to increase the expression of endogenous AHL synthase LuxI, so all underlying modulation of gene expression remains intact, including expression of the *vioABCDE* operon (Figure 4A). A clear pattern of unpigmented cells in a mixed (equal mix of pigmented and unpigmented isolates) colony biofilm model confirmed the inability of the Δ*luxR* mutant to respond through the AHL-LuxI/LuxR pathway (Figure 4B).

It is interesting to note that clear patterns at the edge of mixed colony biofilm indicated zip-like meso-scale structures (Appendix A). The appearance of these structures was similar to recently observed intra-colony channels in *E. coli* [52].

### 3.4. Adaptability to Low Temperatures

Even though *C. vaccinii* was isolated during the cold season, loss of viability was described during *C. violaceum* exposure to low temperatures [53]. During storage of plates (TSA, LB, and blood agar) with isolated *C. vaccinii at* 4 °C, we observed complete loss of bacterial culture recovery from a single colony after several days (within 1 week), while recovery during storage at 25 °C was restricted by drying of solid medium (4 weeks). In the genome of isolated *C. vaccinii*, we revealed some genes responsible for temperature adaptation: ABC transporters for the putrescine import, genes for spermidine synthesis and export, the gene of cold shock protein, and the operon for trehalose transport. To investigate the ability to survive at low temperatures in liquid LB medium, we incubated stationary phase cultures of ChrSV at 4 or at 25 °C for different periods of time. After at least 4 h of incubation at 4 °C, we observed a two-fold decrease in viability in comparison with 25 °C incubation; by 1 week of incubation, this viability difference was more than 20 times lower and by the fourth week of incubation, we observed complete loss of viability for the sample in the 4 °C storage condition (Figure 5a). At the same time, recovery of *C. vaccinii* from initial root-associated biofilm samples placed in the same LB medium was possible at least after 6 months of storage at 4 °C (due to numerous numbers of *C. vaccinii* colonies, we evaluated recovery in a qualitative manner).

Changing membrane fluidity through balancing saturated/unsaturated fatty acid content is an important part of bacterial cold adaptation [54]. To explore the ability to change SFA/MUFA content during exposure to different temperatures, we performed FAME analysis for bacterial culture grown up at 25 °C and then shifted to 4 or 37 °C for 48 h. After 48 h of exposure to 4 °C, we observed a significant shift to monounsaturated fatty acids (MUFAs) and the opposite in the case of 37 °C exposure (Figure 5b). So, isolated *C. vaccinii* has at least one mechanism of low-temperature stress reaction: changing membrane fluidity, but this attempt did not prevent loss of viability in monoculture, while the multi-species environmental biofilm sample provided an opportunity to survive for at least half a year at 4 °C in vitro.

### 3.5. Surrounding Bacterial Community

#### 3.5.1. Culturome Analysis

Bacteria grown on the solid mediums from the root-associated biofilms of *Carex* spp. were defined as culturome. Conventional Biotyper^®^ identification of culturable bacteria was significantly restricted due to the low number of environmental species in the database, so we used the IDBac approach to build a phylogenetic grouping based on MALDI-TOF MS small molecule data using the algorithm described by Laura M. Sanchez and Brian T. Murphy [22,55]. Cooperation in terms of support during growing requires crossing of metabolic pathways between different species. While it is difficult to predict cross-species metabolic interaction in natural multispecies communities, the IDBac approach estimated the small metabolites fingerprint [55]. For the general understanding of possible *C. vaccinii* metabolism crossing with other bacterial species, we created a metabolic-associated network (MAN) with all culturable bacteria within the same biofilm (Figure 6).

The metabolic-associated network demonstrated clear clustering of *C. vaccinii* within culturable bacteria and showed compounds with unique mass (at least 20 masses were not found in any other bacteria within culturome); probably, there were specialized metabolites by *C. vaccinii*. At the same time, a lot of crossing with other species was observed as a sign of integration into the bacterial community. A mass list of the small molecule metabolites is available in Appendix A.

For evaluation of the IDBac approach in bacterial identification, we performed *16S rDNA* sequencing for selected isolates from different groups and confirmed their correct classification at least on a group level. This gave us a general understanding of prevalent bacterial groups in culturome. In root-associated biofilms of *Carex* spp., we found that members of Pseudomonadaceae accounted for more than 80% of the culturable bacterial species (Figure 7a).

#### 3.5.2. Microbiome Analysis

For description of the whole microbial community within the root-associated biofilm, we performed massive parallel sequencing of *16S rDNA* amplicons. Microbiome was presented by the eight phyla: Acidobacteria, Actinobacteria, Armatimonadetes, Bacteroidetes, Cyanobacteria, Firmicutes, Nitrospirae, and Proteobacteria (Figure 7b). The most abundant were Firmicutes (75.7%), followed by Proteobacteria (18.6%). Firmicutes included the five genera, among which *Clostridia* predominated (99.9%). *Clostridia* revealed in the root-associated microbiome were phylogenetically most closely related to a psychrophilic species *Clostridium estertheticum* and other species of the cluster I clostridia isolated from an Antarctic microbial mat [56]. Proteobacteria were presented by the 5 classes, including 73 genera (Figure 8).

The most abundant were Alfa- and Gammaproteobacteria, but the taxonomic diversity of Deltaproteobacteria was as great as Gammaproteobacteria. *Pseudomonas* predominated in Gammaproteobacteria (64%), which was partly consistent with the data of the culturome. However, *Serratia* was revealed only in a trace amount, the same as *Chromobacterium*. It should be noted that in the microbiome, few abundant Betaproteobacteria were represented mainly by the order Burkholderiales and only one genus of Neisseriales was detected. The amount of *Serratia* and *Chromobacterium* revealed in the microbiome is in contrast to the fact that these genera did appear in culturome.

The abundance of the phylum Acidobacteria (2.6%) was an order of magnitude lower than Firmicutes and Proteobacteria but agreed with the data of Pankratov et al. on the quantification by FISH of acidobacteria in native peat sampled from sphagnum-dominated wetlands of different geographic locations (0.1–4.1%) [57]. The phylum Bacteroidetes was even less abundant (1.8%). Additionally, as demonstrated by Li et al., the presence of Bacteroides in the root-associated biofilm may depend on the type of the plant: Bacteroides were detected in samples of narrow-leaved cattail roots but were absent in common reed root samples [58].

Analyzing 20 of the most abundant genera, we found they belong to the next phyla: 1—Firmicutes, 10- Proteobacteria, 2—Bacteroidetes, 5-Acidobacteria, 1—Actinobacteria, and 1—Armatimonadetes. Thus, the microbial community of the root-associated biofilm demonstrated amazing diversity even in the cold winter season.

## 4. Discussion

*Chromobacterium* species demonstrate significant biological activity against other microbes and insects in laboratory settings, but little is known about the overall context of their natural habits and their environmental adaptation. They are widely distributed in tropical and subtropical zones and still believed to be tropical bacterial species. Investigation of environmental cases of *Chromobacterium* species isolation in the temperate climate zone and analysis of their adaptation potential, surrounding microbiome, and description of their natural niche may shed light on the system biology of complex environmental communities and microbial biogeography at a time of discussion around global warming.

We described a case of a resident *C. vaccinii* in root-associated biofilm in a quaking bog in the Dfb climate zone. Nowadays, there are numerous cases of environmental *Chromobacterium* isolation in Europe and one of them is also related to bog, which is in line with the first reported case of *C. vaccinii* isolation from cranberry bogs in Cape Code, USA [3,7]. Additionally, the ability to grow in vitro in low salinity and the water composition from the site of sample collection support the existence of *C. vaccinii* in an oligotrophic environment. Biofilm formation in an environment with low levels of nutrients and significant climate variations around the year helps to create a sustainable surrounding due to the accumulation of nutrients in the biofilm matrix and create a network for metabolic cooperation to degrade xenobiotics and enhance resistance to threats [59].

In support of the unique biofilm mode of cooperation, we found a QS-deficient mutant among *C. vaccinii* isolates, which has a mutation in the key regulatory system LuxI/LuxR. The QS-deficient mutant had no visible pigmentation due to an inability to respond and amplify AHL signaling essential for vioABCDE operon induction. This observation is in line with recently published data: violacein biosynthesis depends on the LuxI/LuxR quorum sensing system and as was recently published, the *ΔcviR* (homolog of *luxR*) mutant of *C. violaceum* lost visible pigmentation due to a dramatic drop in violacein production [48]. Absence of pigmentation is the most noticeable phenotype of *ΔluxR*, while it is well-known that an inability to respond through the LuxI/LuxR system leads to significant changes in other important processes and minimization of the cooperative traits. Interruption of QS-mediated regulation makes crosstalk with other bacteria complicated and might lead to minimization of production but not consumption of public goods [60]. Occurrence and persistence of quorum-sensing bacteria represent social cheaters within the biofilm and stress the general idea of public goods in the bacterial social community. It is important to note that in our study, the QS-deficient mutant was isolated from a natural complex biofilm so this is an additional point to support the laboratory-proved theory around social cheaters in biofilms. One of the costly social goods might be polysaccharides or other molecules for bacterial coating and providing protection from temperature perturbations. This could explain the viability of tropical *Chromobacterium* during the cold winter season.

While isolated *C. vaccinii* showed in vitro some adaptability to sub-zero temperatures through changes of the membrane fatty acid saturation, after long-term storage, recovery was dramatically better from complex initial biofilm samples rather than from pure bacterial suspension. Such cold resistance in a prism of the biofilm lifestyle might be related to the extracellular matrix composition, which serves as a social good [61]. This is consistent with the previous observations that environmental parameters rather than phylogeny determine the composition of biofilm matrix in microbial mats from extreme environments [62]. It might be proposed that the ability of *Chromobacterium* to survive in the bog during the cold winter season was due to sharing goods provided by a diverse microbial community of the root-associated biofilm of *Carex* spp.

Analysis of the surrounding bacterial community by *16S rDNA* microbiome showed a prevalence of psychrophilic anaerobic *Clostridium* species, previously described as members of a microbial mat in an Antarctic freshwater lake [56]. The second prevalent bacterial belong to the *Rhodoblastus* genus, which is a freshwater bacteria and might be associated with sphagnum peat [63,64]. The vast majority of other bacteria are also known as psychrophilic or psychrotolerant. Besides the psychrophilic clostridia (Firmicutes) mentioned above, Acidobacteria are adapted to growth at low temperatures, as demonstrated Pankratov et al., who isolated the acidobacteria from nine Sphagnum-dominated wetlands of West Siberia and European North Russia [57]. *Steroidobacter agariperforans* (Gammaproteobacteria, Nevskiales) may be characterized as psychrophilic so far as ATCC recommended growing the type strain BAA2459 at 3 °C (https://www.lgcstandards-atcc.org/). Another representative of Nevskiales, *Povalibacter*, was revealed with a high abundance from natural and constructed wetlands, demonstrating stability in different geographic zones [65]. From the two genera of Bacteroides, *Mucilaginibacter* was characterized as psychrotolerant by Pankratov et al., who isolated this bacterium from the phagnum peat bog Bakchar, in the Tomsk region of western Siberia [66], and *Flavobacterium* was described as “cool to cold environments” by Van Trappe et al., who investigated bacteria from Antarctic lakes [67].

The four genera of Proteobacteria belong to methanotrophic bacteria, which could be characterized as psychrotrophs (facultative psychrophiles, or psychrotolerants) according to the data of Kevbrina et al., who demonstrated that some methanotrophic species could grow at 10 °C [68]. *Silvanigrella aquatica* (Oligoflexia) can be recognized as psychrotrophs too on the basis of growth in culture in the temperature range of 10–32 °C and the habitat in the freshwater lake located in the Black Forest Mountains (Schwarzwald), Germany [69]. *Pseudomonas*, the third abundant genus in the biofilm community, is known to be cold tolerant due to a wide geographic distribution and ability to grow even at a high-altitude location in the northwestern Indian Himalayas [70].

However, in relation to some bacteria from the list of the most abundant in the root-associated biofilm, the same question arises as in relation to *Chromobacterium*: how can a bacterium that has a temperature optimum in a monoculture at 25–30 °C and quickly dies at 4 °C accumulate in large quantities at low temperatures? If *Fimbriimonas* (Armatimonadetes) is mesophilic with a growth temperature range of 15–30 °C [71], then *Rhodoblastus* (Alphaproteobacteria) was described as mesophilic with optimum growth at 25–30 °C, when cultivated in laboratory, but it has been isolated from the Sosvyatskoe ombrotrophic bog located in Tver Region with cold winter temperatures [72]. Moreover, the type strain IC-180T of the genus *Aciditerrimonas* has growth temperatures of 35–58 °C [73], but at the same time, *Aciditerrimonas* is known as the abundant genus of Actinobacteria in a worldwide range of samples from wetland soil and sediment according to the *16S rDNA* microbiome analysis [74], and Oloo et al. revealed that sphagnum interstitial water samples were enriched in genera, such as *Aciditerrimonas*, on the base of *16S rDNA* sequencing data too [75].

The most intriguing was *Ehrlichia*, which is granulocytic ehrlichiosis agent of humans and other vertebrates, and a tick-borne pathogen. The abundance of *Ehrlichia* was comparable with the abundance of some Acidobacteria and methanotrophic bacteria in the microbial community of the root-associated biofilm. Since culturing *Ehrlichia* species requires a canine macrophage cell line or tick cell line, it is complicated by temperature (34–37 °C) and aerobic conditions for eukaryotic cell growth [76]. All identification of *Ehrlichia* in the environment is associated with ticks. As revealed by Zintl et al., tick density in marsh/bog sites was even slightly higher than in woodland sites [77]. The appearance of *Ehrlichia* in the Moscow region bog could be connected with the change in the distribution of ticks and with the fact that bog is a comfortable place for prolonged nonparasitic phases of ticks, requiring a microclimatic relative humidity of at least 80% to avoid fatal desiccation [78].

Such diverse and complex cultivating conditions of representatives of the root-associated biofilm members support the idea of cooperation within the microbial community to provide a survival opportunity for a broad range of requirements.

Disagreement between the IDBac approach and sequencing data was expected and might be partially explained by the unculturable state of some species and inappropriate culture conditions for growth, like the aerobic condition, inappropriate temperature, and medium composition. Meanwhile, we noted an unexpected prevalence of some species among colonies on solid medium despite their low abundance according to NGS data: *Serratia* accounted for less than 0.003% from all *16S rDNA* bacteriome and was presented >1500 times lower than *Pseudomonas spp.* but still grew and was randomly picked for low-throughput IDBac analysis. The same is true for the *Chromobacterium* genus. A significant limitation in culturome data collection is not only well-known growth competition but also growth cooperation in favorable conditions for some species in closer proximity on the Petri dish in laboratory settings.

Thus, the combination of microbiological and genomic approaches provides a versatile understanding of the microbial community of the root-associated biofilm.

## 5. Conclusions

Altogether, we described the isolation of resident *C. vaccinii* from an environmental complex biofilm in a temperate climate zone, which is not common for members of the *Chromobacterium* genus. This *C. vaccinii* has several genotype and phenotype unique properties in comparison with all other members of the genus. Additionally, an unpigmented isolate with interrupted QS-mediated signaling was represented by social cheaters within the biofilm and might be a sign of adaptation to the community lifestyle through minimization of costly production of social goods, while survival during exposure to sub-zero temperatures (i.e., winter season) completely relies on the surrounding microbial community and factor serving as a sharing good produced by other biofilm members will be in focus for the future research.

## Figures and Tables

**Figure 1 microorganisms-08-01696-f001:**
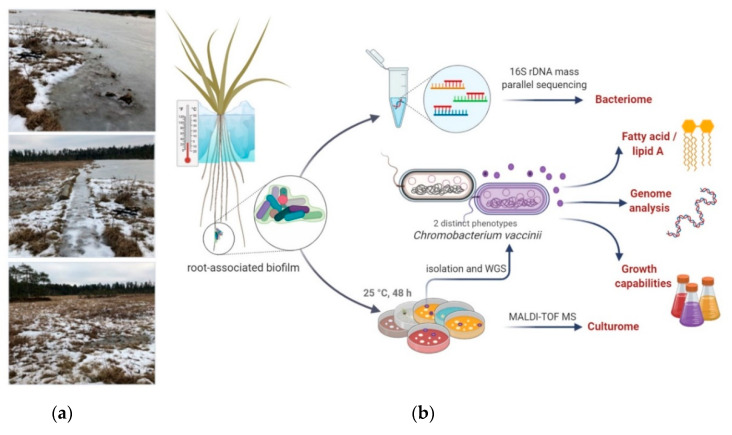
General description of the study design. (**a**) Photography of the location at the day of sampling. (**b**) Sample processing workflow. Picture was created with BioRender.com.

**Figure 2 microorganisms-08-01696-f002:**
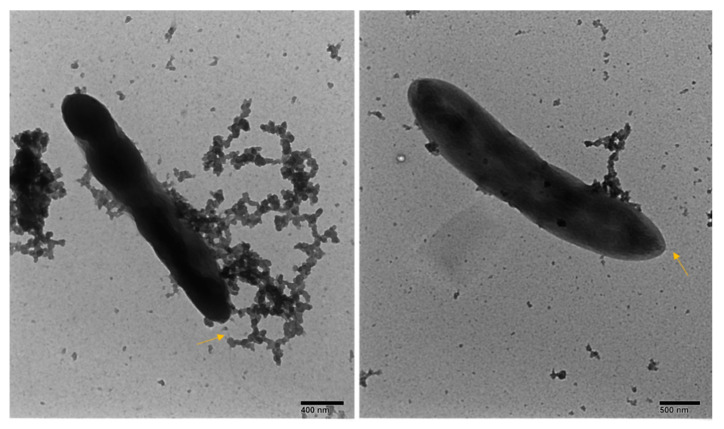
Negative staining electron microphotographs of isolated *C. vaccinii*-pigmented strain (**left**) and unpigmented strain (**right**). The arrow shows the flagellum of bacteria.

**Figure 3 microorganisms-08-01696-f003:**
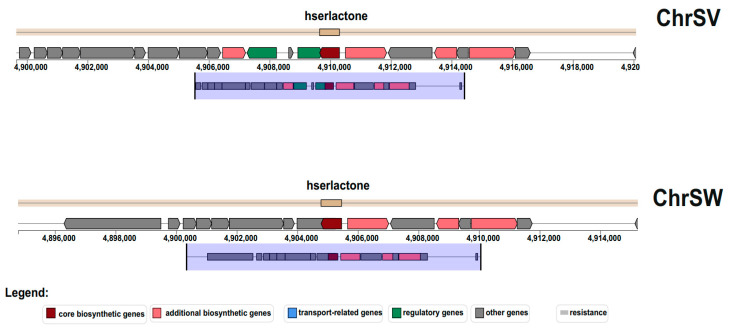
Biosynthetic gene cluster for homoserine lactone in natural pigmented (ChrSV) and unpigmented (ChrSW) isolate of *C. vaccinii.* The resource antiSMASH v.5.1.2 was used for prediction.

**Figure 4 microorganisms-08-01696-f004:**
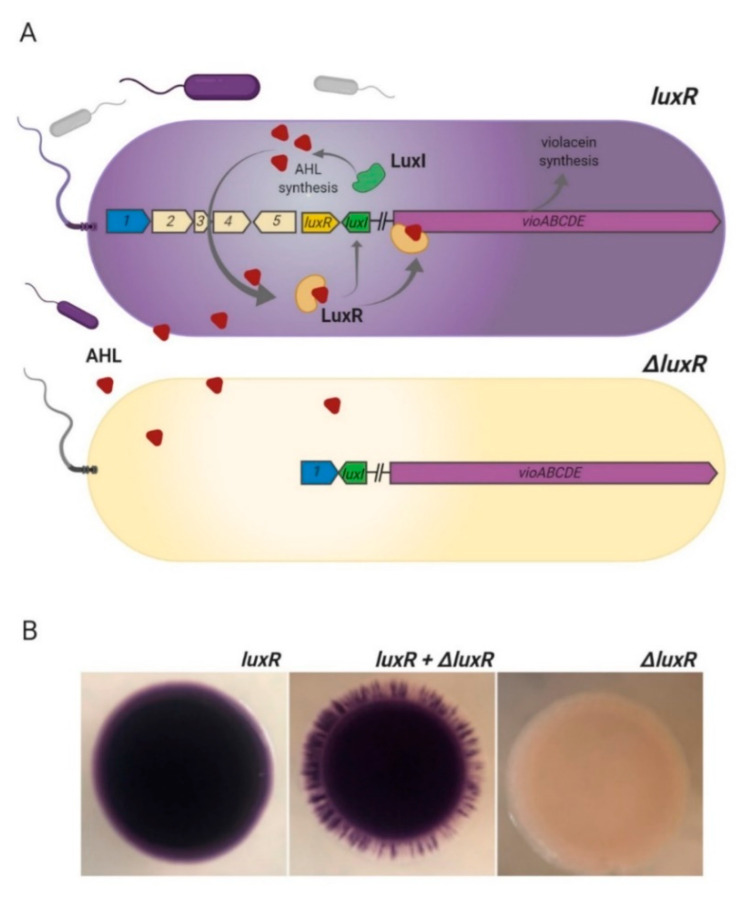
Nature of the unpigmented strain. (**A**) Mechanism of luxI/luxR regulation of violacein production and inability to develop pigmentation in natural *ΔluxR* isolate. LuxR binds with exogenous AHL and activates LuxI production; LuxI syntheses endogenous AHL; increasing AHL concentration amplify AHL-QS signaling loop; increasing concentration of LuxR-AHL complex activates the vioABCDE operon and bacteria produce violacein. In case of the *ΔluxR* strain, AHL signaling is interrupted and vioABCDE is not active. Unpigmented natural isolate of *C. vaccinii* has deletion of luxR and 1-5 protein-coding genes. (**B**) Inability of natural *ΔluxR* isolate to produce violacein in a mixed biofilm colony model. Biofilm colonies formed by pigmented strain (*luxR)*; mix of pigmented and unpigmented strains *(luxR + ΔluxR);* unpigmented strain (*ΔluxR).* Mixed biofilm colony has clear patterns of pigmented and unpigmented zones, which indicates the inability of *ΔluxR* strain to respond and amplify AHL-QS signal and produce violacein in a mixed bacterial population. Picture was created with BioRender.com.

**Figure 5 microorganisms-08-01696-f005:**
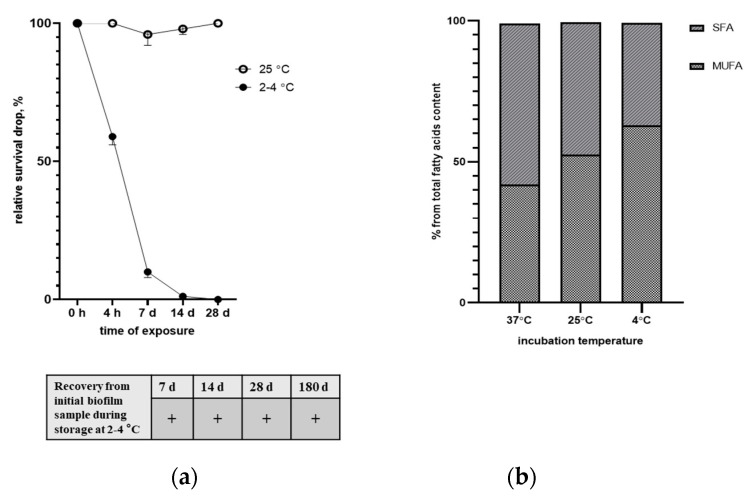
Influence of low temperatures on *C. vaccinii*. (**a**) Survival under exposure of liquid culture of *C. vaccinii* to sub-zero (2–4 °C) temperatures presented in %% relative exposure to 25 °C (presented as 100%) and recovery from the initial biofilm sample during storage at 2–4 °C. (**b**) Membrane fluidity adaptation through changing saturation of fatty acids during exposure to different temperatures.

**Figure 6 microorganisms-08-01696-f006:**
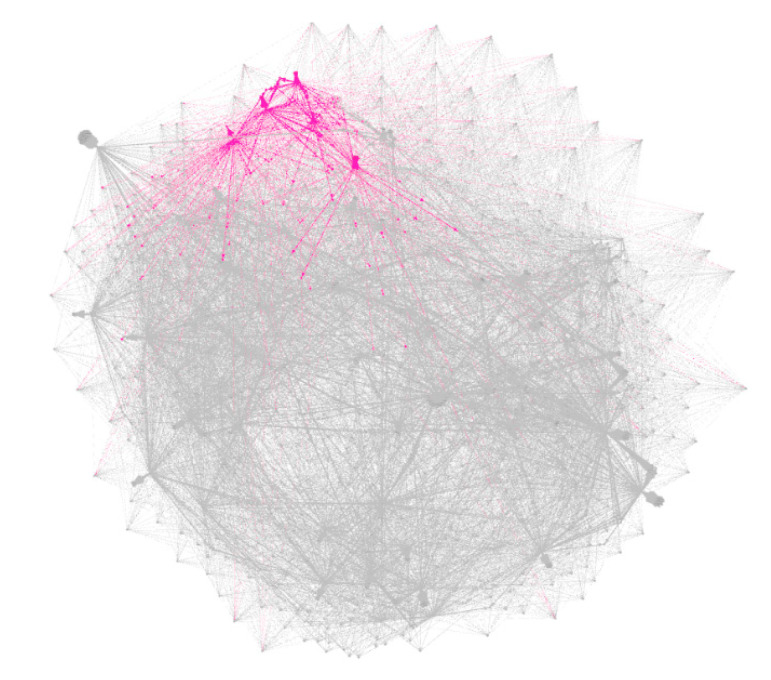
Metabolic-associated network (MAN) of all culturable bacteria from the same with *Chromobacterium* biofilms. Small molecule metabolites from *C. vaccinii* are colored in pink, metabolites from other species are colored in grey. Cumulative distance was built via modularity analysis with default thresholds in Gephi after matrix and media signals were subtracted automatically from the network in IDBac.

**Figure 7 microorganisms-08-01696-f007:**
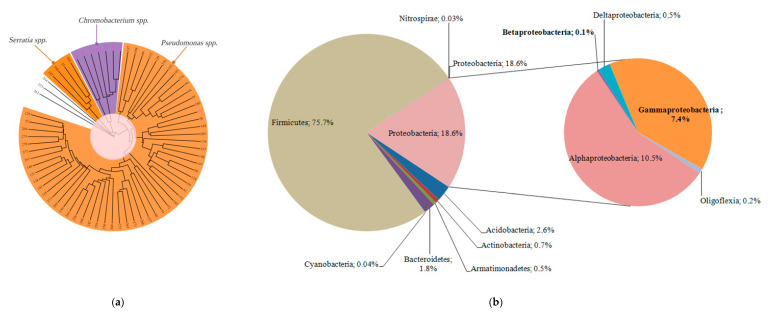
Surrounding bacterial community in biofilm. (**a**) Culturome of root-associated biofilm obtained through IDBac fingerprint grouping. Numbers represent the colony identification number on plates. Mixed species colonies excluded from analysis; (**b**) Proportion of different bacterial taxonomic groups in *16S rDNA* microbiome of the root-associated biofilm. Inset in the diagram reveals the phylum of *Proteobacteria.*

**Figure 8 microorganisms-08-01696-f008:**
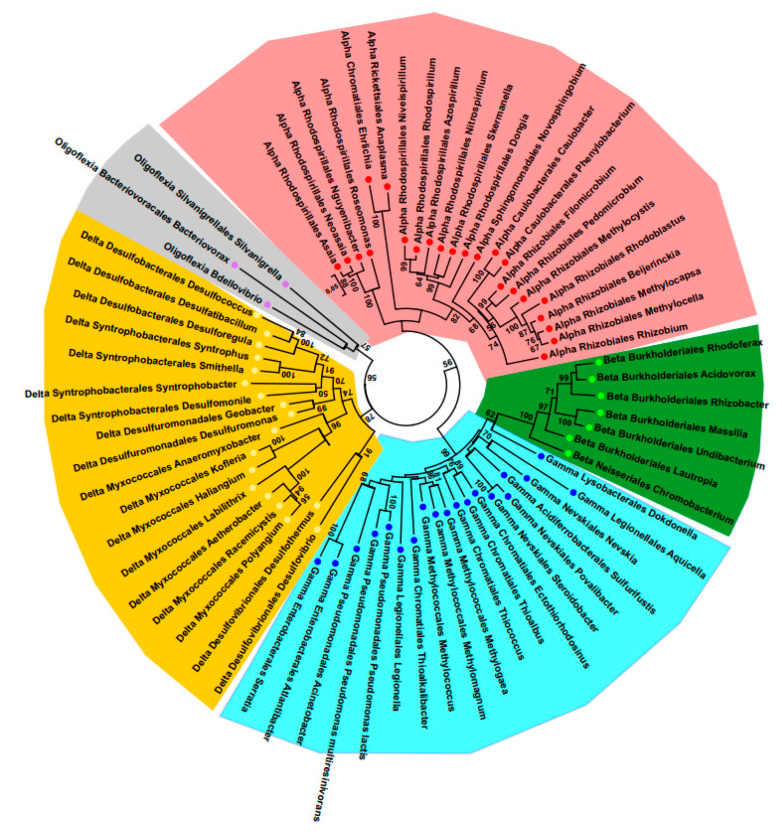
Taxonomic diversity of Proteobacteria in the microbiome of the root-associated biofilm. Red circles—Alphaproteobacteria, blue circles—Gammaproteobacteria, yellow circles—Deltaproteobacteria, green circles—Betaproteobacteria, purple circles—Oligoflexia.

**Table 1 microorganisms-08-01696-t001:** Water quality analysis.

General Characteristics			Cations			Anions			Heavy Metals		
**Turbidity**	2.37	OD530nm	**Mg^2+^**	0.26	mg/L	**[SO4]-**	1.21	mg/L	**Hg**	<0.00001	mg/L
**Chromaticity**	36.9	°	**Ca^2+^**	0.66	mg/L	**[Cl]-**	1.14	mg/L	**V**	<0.0001	mg/L
**Odour**	0	grade 0–5	**Fe^2+^**	0.154	mg/L	**[NO]-**	0.419	mg/L	**Ba**	0.003	mg/L
**pH**	5.68	pH units	**K^+^**	0.43	mg/L	**[HCO3]-**	<6.1	mg/L	**Be**	<0.0001	mg/L
**Hardness**	<0.060	mg-CaCO3/L	**Na^+^**	0.66	mg/L	**[CO3]2-**	<6.0	mg/L	**B**	<0.05	mg/L
**Chemical oxygen demand**	22.2	mg/L	**Al3^+^**	0.051	mg/L	**[NO2]-**	<0.1	mg/L	**Mo**	<0.0001	mg/L
**H2S**	<0.002	mg/L	**[NH4]^+^**	0.38	mg/L	**[Br]-**	<0.05	mg/L	**Co**	<0.0001	mg/L
**Petroleum products**	0.048	mg/L	**Li^+^**	<0.001	mg/L	**[PO3]-**	<0.1	mg/L	**Ag**	<0.0001	mg/L
**Free alkalinity**	<0.1	mM/L				**[F]-**	0.159	mg/L	**Zn**	0.01	mg/L
**Total alkalinity**	<0.1	mM/L							**Ni**	<0.0001	mg/L
**Sulfide minerals**	<0.002	mg/L							**Si**	0.556	mg/L
**Dry weight**	7.37	mg/L							**Cr**	<0.0001	mg/L
**Conductivity**	11	mkS/sm							**Sr**	0.003	mg/L
									**Cd**	<0.0001	mg/L
									**As**	<0.0001	mg/L
									**Cu**	0.002	mg/L
									**Pb**	<0.0001	mg/L

**Table 2 microorganisms-08-01696-t002:** Biochemical properties of *C. vaccinii* isolated strains.

**Characteristic**	Fermentation or Number of +/− Isolates if Variable
oxidase	−
indole production	−
bGL	−
NAG	16/18
SCI	16/18
LAC	−
MAN	1/18
TRE	+
XYL	5/18
ARA	3/18
aGA	−
bGA	−
MAL	−
GAL	1/18
MLT	−
CEL	−
SUC	2/18
INO	−
GGT	+
PHS	+
ESL	−
H2S	−
MAL	−
ONP	−
SAL	−
SOR	−
MLB	−
GLP	−
DUL	−
ADO	−
ART	−
RAF	−
bXY	−
NaCl, % range	0–3
pH range	4.0–8.0
pigmentation	3/18

+ = all tested isolates were positive; − = all tested isolates were negative; numbers represent positive isolates from total number of tested isolates.

**Table 3 microorganisms-08-01696-t003:** Fatty acid compositions of *Chromobacterium* species.

	1	2	3	4	5	6	7	8	9	10	11	12	13	14
11:0	−	−	0.2	−	−	−	−	−	−	0.2	−	−	−	−
10:0 3OH	1.8	3.2	3.4	4.3	4.7	2.9	3.7	1.5	3.2	4.6	3	2.2	5.1	2.4
12:0	2.4	3.8	3.8	5.0	3.9	4.0	4.2	9.7	3.3	8.8	3.2	3.1	4.9	3.3
11:0 3OH	−	−	−	−	−	−	−	−	−	−	−	−	0.4	−
13:0	−	−	−	−	−	−	−	−	−	−	−	−	0.4	−
12:0 2OH	1.8	1.9	2.0	2.9	2.4	1.4	2.3	−	1.9	0.2	1.9	1.6	3.3	1.7
12:0 3OH	2.8	3.3	3.4	4.0	3.6	2.5	2.9	1.4	2.9	4.4	2.8	2.5	4.8	2.6
13:0 2OH	−	−	−	−	−	−	−	0.4	−	−	−	−	−	−
14:1 w7c	0.4	−	−	−	−	−	−	−	−	−	−	−	−	−
14:1 w5c	0.1	0.2	0.2	0	0.4	−	0.3	0.4	−	−	0	0.2	−	−
14:0	2.2	2.3	2.1	3.2	2.5	2.3	3.3	4.0	2	2.6	3.1	2.5	3.5	2.0
15:0 iso	−	−	−	−	−	−	−	0.5	−	−	−	−	−	−
15:0 iso G	−	−	−	−	−	−	−	0.7	−	−	−	−	−	−
15:1 w8c	0.1	−	−	−	−	−	−	−	−	−	−	−	−	−
15:1 w6c	0.2	−	−	−	−	−	−	−	−	−	−	−	−	−
15:0	1.0	−	−	1.3	−	−	0.9		1	0.6	2.3	−	3.0	−
16:1 w7c	43.7	42.7	41.9	41.9	47.1	42.5	34.1	38.6	38.9	33.4	28.7	38.5	27.5	36.3
16:1 w5c	0.4	0.3	0.3	0.3	0.5	−	−	−	−	0.3	0	0.2	−	−
16:0	28.4	28.4	29.6	25.0	24.0	27.3	26.1	29.7	30.2	25.8	32	31.5	26.6	28.5
17:1 w6c	−	−	−	0.2	−	−	0	−	0.2	−	−		0.4	−
17:0 CYCLO	0.2	0.4	−	−	0.4	−	2.9	−	−	−	13.2	0.2	4.3	1.3
18:2 w6,9c	−	−	−	−	−	−	4.2	−	−	−	−	−	−	−
18:1 w9c	−	−	−	−	−	−	2	−	−	−	−	−	−	−
18:1 w7c/12t/9t	12.5	13.1	12.6	10.6	10.3	12.0	12.3	5.5	15.7	18.8	8.7	15.9	14.8	19.3
18:0	1.8	0.4	0.5	−	0.2	0.6	0.4	1.6	0.5	0	0.6	0.5	0.3	0.4
SFA/MUFA *	0.6	0.6	0.7	0.7	0.5	0.6	0.7	1.0	0.7	0.7	1.1	0.7	0.8	0.6
Hydroxy FA **	6.4	8.4	8.8	11.2	10.7	6.8	8.6	3.3	8	9.2	7.7	6.3	13.6	6.7

1, GIMC1602:ChrSima_v strain in this study; 2, MWU300-*C. vaccinii* [7]; 3, MWU205-*C. vaccinii* [7]; 4, PRAA4-1-*C. subtsugae* [42]; 5, IIBBL 14B-1 *C. sphagni* [6]; 6, CCUG 53,230 *C. haemolyticum* [43]; 7, ATCC 12,472 *C. violaceum* [42]; 8, LAM1188 *C. rhizoryzae* [8]; 9, DSM 170,043-*C. subtsugae* [42]; 10, CC-SEYA-1-*C. aquaticum* [44]; 11, LMG 3947- *C. piscinae* [45]; 12, IIBBL 112-1 *C. phragmitis* [46]; 13, LMG 3953 C. *pseudoviolaceum* [45]; 14, CBMAI 310T *C. amazonense* [47]. * saturated FA/monounsaturated FA; ** summed hydroxy fatty acids.

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
