# Peer review of "Integrated into Environmental Biofilm Chromobacterium vaccinii Survives Winter with Support of Bacterial Community"

_microorganisms, 2020, doi:10.3390/microorganisms8111696_

Round 1

Reviewer 1 Report

Dear authors,

I carefully read your manuscript entitled in “Integrated into environmental biofilm Chromobacterium vaccinii survives winter with support of bacterial community”. Your research groups found tropical bacterium from Moscow and the characterized your strains. Your results are interesting in not only microbial physiology but also microbial ecology. However, I think that your manuscript includes some of verbosities. Because, your core results are figs 5 and 8 related with response C. vaccinia under low temperature. If your manuscript is submitted in the journal based on paper publishing style, I recommend to move the results of bacterial diversity of roots associated bacteria to the supplemental material.

This manuscript still remained some of mistakes in units such as “sm” (cm ?), ÌŠF (it is not SI unit), only “C” (is not meaning of temperature). “sp.” and “spp.” are not italic, respectively.

Most of the problem was that figure 2 was absent. Therefore, I did not recommend to publish this manuscript in this journal. Authors should resubmit revised manuscript.

Author Response

Point 1: However, I think that your manuscript includes some of verbosities.

Dear reviewer, thank you for this feedback – we have reduced some description of genetic features and description of OMVs.

Point 2: Because your core results are figs 5 and 8 related with response C. vaccinia under low temperature.

Response 2: Dear reviewer, thank you for pointing this, after your comments we have rethought about sequence of our results and have moved figure 8 at the beginning of 3.5 section (Surrounding bacterial community) as well as its description and discussion.

Point 3:  I recommend to move the results of bacterial diversity of roots associated bacteria to the supplemental material

Response 3: Thank you for this suggestion, but we believe in importance of microbial community and its significant impact in adaptation to environmental threats, that’s why we have modified description of root-associated microbiome, but still leave it as main results.

Point 4: This manuscript still remained some of mistakes in units such as “sm” (cm ?), ÌŠF (it is not SI unit), only “C” (is not meaning of temperature). “sp.” and “spp.” are not italic, respectively.

Response 4: Thank you for your careful review – we have fixed the issue

Point 5: Most of the problem was that figure 2 was absent. Therefore, I did not recommend to publish this manuscript in this journal. Authors should resubmit revised manuscript.

Response 5: Sorry for this, but we checked documents right after submission and all figures were at their place. Probable it was a technical problem because right after submission figure 2 was. For the resubmission we have checked carefully all figures and have drown editor`s attention in this issue.

Reviewer 2 Report

I enjoyed the perspective of the study from a climate change point of view, and the phenotypic and genomic characterization of the C. vaccinii strains. The methods and conclusions are justified and the information is important. However, the manuscript needs extensive language edition as a lot of the main messages are not clearly presented, particularly the Results section. See comments below:

Abstract

Lines 14-15: This sentence does not make sense. Did the authors mean  “Chromobacterium species are common in tropical and subtropical zones in environmental water samples and believed to belong to «tropical» species”?

Introduction:

Line 36: Replace “: pollution, climate change…” with “including pollution, climate change…” and throughout the text where authors use “:” (e.g. line 58)

Line 40: Did the authors mean “unexplored” instead of “pure explored”

Lines 40-42 may not be needed as they do not add to the main idea.

Materials and Methods

Line 74: Does Carex spp. have a common name? Please add.  Spp. should not be italics. Please modify.

Line 77: Please remove “for microbiology observation”

Line 84: Add degree sign before C in “25 C” and throughout the manuscript as needed (e.g., line 237).

Line 89: Please combined both references so they are both inside the brackets (i.e., [20,21])

Results

Since Figure 1 is summarizing the methods, please move to the Materials and Methods section and reference in the text appropriately.

Line 222: Modify sentence “Quarry Sima used to exploitation of peat in 18-19 centuries” to “Quarry Sima was used for the exploitation of peat in the 18-19 centuries”

Line 235: Replace “at” with “during”

Line 271: Did the authors mean “contents” instead of “contains”?

Line 287: Can the authors clarify what they mean with “using modified for pigmented strains method”?

Lines 285-299: May need language edition.

Lines 322-351: May need language edition.

Line 536: Did the authors mean site instead of cite?

Line 541: Did the authors mean sign instead of sing?

Author Response

Response to Reviewer 2 Comments

Dear reviewer, thank you for your careful review and detailed suggestions – that let us significantly improve our manuscript. We regret there were problems with language. The paper has been carefully revised to improve the grammar and readability.

Point 1:

Introduction:

Line 36: Replace “: pollution, climate change…” with “including pollution, climate change…” and throughout the text where authors use “:” (e.g. line 58)

Has been corrected

Line 40: Did the authors mean “unexplored” instead of “pure explored”

Has been corrected

Lines 40-42 may not be needed as they do not add to the main idea.

While we appreciate the reviewer’s feedback, we respectfully disagree. Thank you for this suggestion, but we would like to leave this sentence because usually clinical cases are better described in details rather than environmental cases and we believe in importance of stress this difference for further research.

Materials and Methods

Line 74: Does Carex spp. have a common name? Please add.  Spp. should not be italics. Please modify.

We have added common name «sedge»

Line 77: Please remove “for microbiology observation”

Has been corrected

Line 84: Add degree sign before C in “25 C” and throughout the manuscript as needed (e.g., line 237).

Has been corrected

Line 89: Please combined both references so they are both inside the brackets (i.e., [20,21])

Has been corrected

Results

Since Figure 1 is summarizing the methods, please move to the Materials and Methods section and reference in the text appropriately.

We think this is an excellent suggestion and have moved the figure and reference up.

Line 222: Modify sentence “Quarry Sima used to exploitation of peat in 18-19 centuries” to “Quarry Sima was used for the exploitation of peat in the 18-19 centuries”

Has been corrected

Line 235: Replace “at” with “during”

Has been corrected

Line 271: Did the authors mean “contents” instead of “contains”?

Has been corrected

Line 287: Can the authors clarify what they mean with “using modified for pigmented strains method”?

Thank you for pointing out – we have added clarification in materials and methods section. Also, we have improved discussion with hypothetical explanation of the negative oxidase test results.

Lines 285-299: May need language edition.

Has been edited

Lines 322-351: May need language edition.

Has been edited

Line 536: Did the authors mean site instead of cite?

Has been corrected

Line 541: Did the authors mean sign instead of sing?

Has been corrected

Round 2

Reviewer 1 Report

Dear authors,

I am satisfied with revised manuscript and recommend to accept in this journal.

Reviewer 2 Report

The authors have addressed the concerns and questions presented. The experimental design and analysis methods seem appropriate.